# Biomimetic Octacalcium Phosphate Bone Has Superior Bone Regeneration Ability Compared to Xenogeneic or Synthetic Bone

**DOI:** 10.3390/ma14185300

**Published:** 2021-09-14

**Authors:** Jooseong Kim, Sukyoung Kim, Inhwan Song

**Affiliations:** 1Department of Biomedical Engineering, Yeungnam University, 170 Hyeonchung-ro, Nam-gu, Daegu 42415, Korea; joorpediem@gmail.com; 2School of Materials Science and Engineering, Yeungnam University, 280 Daehak-ro, Gyeongsan 38541, Korea; sykim@ynu.ac.kr; 3Department of Anatomy, College of Medicine, Yeungnam University, 170 Hyeonchung-ro, Nam-gu, Daegu 42415, Korea

**Keywords:** octacalcium phosphate (OCP), hydroxyapatite (HA), biphasic calcium phosphate (BCP), xenogenic bone, synthetic bone, bone regeneration, phase conversion, mineralization

## Abstract

Octacalcium phosphate (OCP) is a precursor of biological apatite crystals that has attracted attention as a possible bone substitute. On the other hand, few studies have examined this material at the experimental level due to the limitations on OCP mass production. Recently, mass production technology of OCP was developed, and the launch of OCP bone substitutes is occurring. In this study, the bone regeneration capacity of OCP products was compared with two of the most clinically used materials: heat-treated bovine bone (BHA) and sintered biphasic calcium phosphate (BCP). Twelve rabbits were used, and defects in each tibia were filled with OCP, BHA, BCP, and left unfilled as control (CON). The tibias were harvested at 4 and 12 weeks, and 15 μm slides were prepared using the diamond grinding method after being embedded in resin. Histological and histomorphometric analyses were performed to evaluate the bone regeneration ability and mechanism. The OCP showed significantly higher resorption and new bone formation in both periods analysed (*p* < 0.05). Overall, OCP bone substitutes can enhance bone regeneration significantly by activating osteoblasts and a rapid phase transition of OCP crystals to biological apatite crystals (mineralization), as well as providing additional space for new bone formation by rapid resorption.

## 1. Introduction

Autogenous bone is still the best option for regenerating bone defects because it meets all the indicated requirements of an ideal bone graft material for osteoinduction and osteoconduction [1,2,3]. On the other hand, the use of autografts is limited by the insufficient bone volume, specific surgical complications, postoperative morbidity, and operation cost. Thus, synthetic bone or biomimetic bone materials as alternatives to autogenous bone have been evaluated as artificial bone in orthopedics and dentistry. These alternative materials should provide a variety of shapes and sizes with mechanical strength and biocompatibility suitable for use in the regeneration of bone defect sites. Generally, bioresorbable materials are preferred because they are expected to maintain the bone volume during bone reconstruction and be gradually replaced by newly formed bone [4]. Numerous physicochemical features of scaffolds, such as surface chemistry, surface roughness, topography, mechanical properties, and interfacial free energy (hydrophobic/hydrophilic balance) are important for cell attachment, proliferation, and differentiation. These factors are also critical to the overall biocompatibility and bioactivity of a particular material [5,6,7].

The resorption of bone substitutes is related to several factors, such as particle size, porosity, chemical and crystallographic properties (composition, Ca/P ratio, phases, and crystallinity), and pH (appropriate for body fluids) [8,9]. In general, smaller particle size, higher porosity, lower crystallinity, and higher non-stoichiometric ratio result in faster resorption from bone substitutes [10]. Hydroxyapatite (HA) and β-tricalcium phosphate (β-TCP) materials are currently used as artificial bone graft materials [11,12]. HA produced by sintering is used as a scaffold material because it does not dissolve in bone defects for a long period of time and retains its shape [13,14,15]. In contrast, β-TCP is used as a resorbable bone substitute due to its inherent solubility in a physiological environment [16]. Biphasic calcium phosphates (BCP), composed of various ratios of HA and β-TCP phases, are mainly applied [17] to control the rate of resorption. Recent study has shown that the resorption of β-TCP can occur not only by dissolution but also by the phagocytosis of osteoclasts [18].

Octacalcium phosphate (OCP), which is attracting attention as an alternative calcium phosphate bone graft material, is biodegradable at the bone defect site and has a neutral pH [19,20]. The OCP crystal, Ca_8_H_2_(PO_4_)_6_ 5H_2_O, has a water layer between two apatite layers. In a physiological environment, the water layer is removed from the OCP, and the two apatite layers are combined to form HA crystals [21]. Based on its crystallographic similarity, OCP has been proposed as a precursor of biological apatite crystals in bones and teeth. Histological studies by Suzuki and coworkers found that some calcium phosphate ceramics, such as β-TCP and OCP, were reabsorbed by osteoclasts in addition to dissolution by the physiological pH of OCP [22,23,24]. The superior osteoconductivity of OCP has also been demonstrated in animal-based studies [4,25].

To date, a few in vivo and clinical studies using OCP have shown that bone repair is superior to other bone substitutes [26,27,28,29]. OCP as a bone substitute has many advantages for bone restoration, but the difficulties in mass production limited its practical clinical application. The recent development of OCP mass production technology has opened up the possibility of the clinical applications of OCP. So far, there has been a laboratory-scale study of OCP substances, but this study is the first comparative animal study of a commercialized OCP product. The bone regeneration ability of OCP products was compared with two of the most clinically used materials: heat-treated bovine bone and sintered BCP.

## 2. Materials and Methods

### 2.1. Bone Substitute Materials

Three commercially available granular products were used in this study. The bone-forming ability of a newly released OCP bone product was compared with two other types of products that are currently widely used. A xenograft product (Bio-Oss), comprised of an inorganic mineralized trabecular bovine HA matrix, was used as a comparative test group. Bio-Oss (Geistlich Pharma AG, Wolhusen Switzerland), which is produced by deproteination at high temperatures, favors the proliferation of blood vessels and bone cell migration through the interconnecting micropores. Another comparative testing sample was a BCP product (MBCP+) that consists of 20:80% of HA and β-TCP. MBCP+ (Biomatlante SAS, Édouard Belin, France) is a BCP synthetic bone graft substitute with a micro- and macroporous structure closely resembling the architecture of natural human bone.

The OCP test product was a granular synthetic OCP material, Bontree (HudenBio, Gwangju, Korea), consisting mainly of OCP. Bontree is a recently released biodegradable synthetic bone graft substitute with a micro- and macro-pore structure. Unlike the above two products, which are processed at high temperatures, Bontree products are produced at room temperature. In general, bone graft materials prepared at low temperature are known to exhibit high bioactivity, but have low mechanical strength and are easily resorbed. In particular, when the OCP material is subjected to high-temperature treatment, it has a characteristic of changing into a different phase.

### 2.2. Characterization of Bone Substitute Materials

The morphology, size, and pore characteristics of the samples were examined by scanning electron microscopy (SEM, Hitachi S-4300; Hitachi, Tokyo, Japan) at ×50 and ×20,000 magnifications. The phase of the bone graft material and the Ca/P ratio directly affect the rate of bone formation and biodegradation. Therefore, the phases of all samples were investigated by X-ray diffraction (XRD, X’pert MPD-PRO; Panalytical, Almelo, Netherlands). The XRD experiments were performed at 40 kV and 30 mA and using copper Kα radiation. The Ca/P ratio of the samples was measured by energy-dispersive X-ray spectroscopy (EDS. Ultim Max; Oxford Instruments, Abindon, UK).

### 2.3. Animals

Twelve eleven-week-old male New Zealand White rabbits were obtained from Orientbio (Seongnam, Korea) and maintained under specific pathogen-free conditions. The Institutional Animal Care and Use Committee (CRONEX-IACUC 201908004) of Cronex Co., Ltd. (Seoul, Korea) approved all experimental protocols. The rabbits underwent initial adaptation while being fed a normal diet under temperature, humidity, and light-controlled conditions. After two weeks of acclimatization, healthy animals within the 80th percentile of body weight (3.0–3.5 kg) were numbered with minimal weight variation and underwent surgery.

### 2.4. Surgical Procedure

The subjects were anaesthetized with an intravenous injection of 5 mg/kg body weight of tiletamine hydrochloride and zolazepam hydrochloride (Zoletil 50; Virbac, Carros, France) and 15 mg/kg body weight of 2% xylazine hydrochloride (Rumpun; Bayer, Seoul, Korea). After a skin incision, both tibias were exposed, and three 3 mm diameter defects were made in each tibia using an implant drill. The defects were filled with Bontree (OCP), Bio-Oss (BHA), or MBCP + (BCP). An additional sham surgery group (CON) was also prepared. After 4 or 12 weeks, the animals were sacrificed, and the tibias were harvested.

### 2.5. Bone Sample Preparation

Each defect segment was cut using a diamond saw and fixed in 10% formalin for one day. The bone samples were dehydrated in a graded series of ethanol and infiltrated for one day each in a 1:3, 1:1, and 3:1 Technovit 7200 resin (Heraeus Kulzer, Wehrheim, Germany) and ethanol mixture. The sample was polymerized in a UV embedding system (Kulzer Exakt, Wehrheim, Germany) after shaking for one day under vacuum in pure Technovit resin. The hard resin sample obtained was cut and ground to 15 μm thick slides using an EXAKT cutting and grinding machine (Kulzer Exakt 300, 400CS; Wehrheim, Germany). The tissue slides were observed by optical microscopy (6000D; Leica, Germany) after being hematoxylin-eosin (H-E) stained and mounted.

### 2.6. Histological Observation and Morphometric Analysis

New bone formation and the implant material status were observed, and the bone healing process for each implant material was analyzed (*n* = 9 each). The new bone formation and the implant material remnant were measured quantitatively using the Image-Pro Plus program (Media Cybernetics, Silver Spring, MD, USA). Two rectangles in a 3 × 1.5 mm size boundary were set on the cortical bone and marrow area in the defect area. An accurate region of regeneration inside a rectangle was gated using the above computer program. The total gated, new bone occupied, and implant material occupied areas were measured based on the color and darkness differences using the Image-Pro program. The percentages of new bone and implant material in the total gated area were calculated.

### 2.7. Statistical Analysis

The data are presented as graphs representing the mean ± standard error. ANOVA (Scheffe) was performed, and a *p* value of <0.05 was considered significant. Statistical analysis was performed using IBM SPSS Statistics 23.0 (IBM, Armonk, NY, USA).

## 3. Results

### 3.1. Bone Substitute Materials

Microstructures evaluated at ×50 magnification of OCP, BHA and BCP revealed that all samples exhibited similar granular sizes with diameters of 1.5–2.0 mm, 1.0–1.7 mm, and 1.0–1.7 mm, respectively. Although the granule shape of each sample was very different due to the difference in the manufacturing process, since the size of the granules used in this study was similar, the size of the macropores formed by gathering these granules was expected to be similar. The shape of the granules in the BHA and BCP samples consisted of irregularly angled granules, whereas the OCP sample consisted of round-shaped granules.

At high magnification (×20,000), the BHA microstructure was in the form of agglomerated nano-sized particles. No growth or crystallization of the grains occurred because it was heat-treated at a lower temperature than the BCP sample. Compared to BHA, the BCP microstructure was heat-treated at a higher temperature, and larger nano-sized rod grains and micropores were well formed because of grain growth or crystallization. Unlike BHA and BCP, the OCP product was prepared at room temperature. An examination at ×20,000 magnification showed that the originally synthesized ribbon-shaped OCP grains had aggregated to form micropores, but no further crystal growth or recrystallization had occurred (Figure 1). The BHA sample had low crystallinity because the heat treatment temperature was not high enough. In contrast, the crystallinity of the BCP sample was very high because of the high-heat treatment. On the other hand, the crystallinity of the OCP sample maintained that of the starting raw material because OCP products were prepared at room temperature. Figure 2 shows the XRD patterns of the samples. XRD phase analysis confirmed that Bontree (OCP) samples consist of OCP phases, whereas Bio-Oss (BHA) and MBCP + (BCP) samples mainly consisted of HA phases and a mixture of HA and β-TCP phases, respectively. XRD phase analysis confirmed that Bontree (OCP) samples consisted of 80% OCP phase, and Bio-Oss (BHA) consisted of 90% HA phase, whereas MBCP + (BCP) samples consisted of 20% HA and 80% β-TCP phases.

The Ca/P ratios of OCP, BHA, and BCP calculated from the EDS results were 1.24, 1.60, and 1.32, respectively (Figure 2). The Ca/P ratio of each of the samples was lower than the stoichiometric ratio (1.33, 1.67, and 1.50) of each material. Hence, each sample is considered to be Ca deficient because the measured Ca/P ratios were lower than the stoichiometric ratios. Because the properties of bone graft materials directly affect the rate of new bone formation and biodegradation, the resulting Ca/P ratio will be very useful when describing the results of animal experiments in the Section 4.

### 3.2. Histomorphometric Analysis

Tibial defect healing progressed in all groups with time (Figure 3). The assessments showed that the CON group had the least cortical bone in the defect area at 4 weeks, but the level was similar to that of the other groups at 12 weeks (Figure 4a). The initial poor bone formation was attributed to a large-sized defect because it is considered challenging to cross 3 mm directly by osteoconduction only. The OCP group showed the most prominent bone formation, and it was the only group to show a significant difference from that in the CON group at four weeks (Figure 4c). Bone formation in the marrow area in the OCP group was an unusual result. Although there was a return to a normal marrow structure at 12 weeks, the OCP group showed a prominent high level of bone in the marrow area at 4 weeks (Figure 4b). New bone formation in the total defected area was significantly high in the OCP group at 4 weeks, maintaining almost the same level at 12 weeks. This was attributed to the offsetting of new bone formation in the cortical area by bone resorption in the medulla area. The BHA group showed a significantly lower level of new bone formation than those of the OCP and BCP groups at 12 weeks (Figure 4c). This is because of a lack of space for bone formation because BHA does not resolve and occupy the space. An analysis of the total amount of bone formation showed that OCP produced the best new bone formation at four weeks, and BHA produced the least amount of cortical bone at 12 weeks (Figure 4c).

Although the implanted material remained as large clumps in the bone marrow site in all implant groups (Figure 3), OCP showed a significantly lower volume and size at 4 and 12 weeks (Figure 4d). In contrast, BHA showed the same level at both periods. BHA and OCP showed the slowest and fastest resorption rate of the implant materials examined (Figure 4c–e).

### 3.3. Histological Findings

Bone healing progresses through the following steps: blood clotting, granulation tissue (procallus), soft callus (fibrocartilage), hard callus (primary bone), and remodeling (mature bone). The 4- and 12-week samples were assumed to be the hard callus and remodeling stages, respectively (Figure 3 and Figure 5).

Four weeks after the OCP implant, well-organized hard callus formation was observed in the cortical and medullary areas (Figure 3a and Figure 5a). At higher magnification, most OCP granules were covered with new bone matrix and osteocytes in the lacuna. Moreover, the new bone surface was covered by numerous osteoblasts, and osteoclasts were observed (Figure 6a,b). The OCP granules that remained in the cortical bone and medulla areas were usually of small size, and there was no evidence of cell infiltration. Bone healing had advanced to a mature form at 12 weeks; the cortical area was filled with mature compact bone with a lamellar structure and a Haversian system; a few small-size traces of OCP granules were observed (Figure 5e and Figure 6c). Several small cavities filled with marrow tissue were observed in the cortical bone, but this was not an OCP-specific feature (Figure 3e,h and Figure 5h). At 12 weeks, the inner marrow area almost returned to the original marrow structure, showing adipocytes and hematopoietic cells. Occasionally, various sizes of OCP remnants covered with thin bone and a capsular structure were observed (Figure 6d).

The BHA group showed a similar healing process to the OCP group, but new bone formation was not as active as that in the OCP group, particularly in the inner medullary area. The BHA granules maintained the original compact bone structure, including lacuna and lamella (Figure 3b and Figure 5b). At 12 weeks, cortical bone was filled compactly with new bone, but the new bone area was smaller than that in the OCP group because the BHA granules still occupied a fair proportion of the space; much more of the area retained soft tissue compared to that in the OCP group (Figure 5f). The marrow area displayed features similar to OCP at 4 weeks, and there were no advances in new bone formation or substrate resorption at that time.

The BCP group showed similar healing patterns to those in the OCP group in the cortical areas, but the BCP granules occupied large portions of the cortical and medulla areas. Moreover, there was no evidence of noticeable resorption (Figure 5c,g).

The CON group exhibited a typical bone healing process. Primary woven bone was observed in the cortical area at 4 weeks, and large holes remained in the central defect area due to incomplete bone healing. Trabecular bone was replaced with well-organized lamellar bone, including evidence of the Haversian system, at 12 weeks (Figure 5d,h). Bone formation was predominantly active on the periosteal side due to the active supply of osteoblasts from the periosteum; the 3 mm defect size was not small enough to allow direct osteoconduction. In some cases, a large marrow space, which included hematopoietic cells, was observed in the cortical bone (Figure 5h).

## 4. Discussion

The solubility or biodegradation of calcium phosphate substances is related directly to the Ca/P molar ratio. Various calcium phosphate materials form differently depending on the Ca/P ratio. A calcium phosphate material with a low Ca/P ratio exhibits chemical stability in a low pH environment, whereas a material formed at a high Ca/P ratio has stability under neutral or basic pH conditions. For example, HA exhibits chemical stability at a high pH (i.e., a neutral or basic environment) because its stoichiometric Ca/P ratio is 1.67. In contrast, OCP has a relatively low Ca/P ratio of 1.33 and exhibits stability at a relatively low pH (i.e., acidic environment). Therefore, in animal experiments, the BHA sample composed mainly of HA would exhibit high chemical stability under physiological conditions. This stability would be followed by BCP (composed of HA and β-TCP) and the OCP sample composed mainly of OCP, which show low chemical stability. Therefore, in a neutral in vivo environment, the OCP sample with a relatively low Ca/P ratio had a much faster biodegradation rate than those of the other samples (Figure 5). These results are consistent with the findings reported elsewhere [24,30].

Acidic calcium phosphates, such as OCP and dicalcium phosphate dihydrate (DCPD), are considered soluble ceramics at a neutral pH. In vivo biodegradation is generally associated with the solubility of calcium phosphate at a physiological pH level [24,30]. In addition, β-TCP is less acidic than OCP but is the most widely used biodegradable ceramic in vivo [19,21], whereas HA is the most chemically stable at physiological pH. In this study, in vivo resorption of the OCP sample, composed mainly of OCP, was approximately two-fold after four weeks and four-fold after 12 weeks, which was notably faster than that of the BCP sample composed of HA and β-TCP and the BHA sample composed of HA. The in vitro resorption rate of OCP was similar to that of β-TCP, but the in vivo resorption of OCP samples was much faster than that of BCP composed of 80% β-TCP. These results show that an additional degradation mechanism is involved in the OCP sample.

In general, the stability of calcium phosphate materials is related to several factors, such as particle size, porosity, Ca/P ratio, phase, and crystallinity [9,10]. Even with the same type of calcium phosphate material, the chemical stability in the physiological environment varies with the degree of non-stoichiometry, particle size, porosity, and crystallinity. Non-stoichiometric materials are thermodynamically and chemically unstable compared to stoichiometric materials [19]. All calcium phosphate bone substitutes examined in this study have non-stoichiometric chemistry: The Ca/P ratio of each tested sample was lower than the stoichiometric ratio. Therefore, all samples used in this experiment are believed to exhibit faster biodegradation than stoichiometric materials.

Micrometer-sized particles take longer to be eliminated than nanometer-sized particles because osteoclasts require a longer time for chemical dissolution and biological absorption. Although the main phase was HA, the BHA samples would have been absorbed quickly because they were comprised of nano-sized particles. SEM (Figure 1) showed that the particle distribution of the BHA samples was in the tens of nanometers range, with a BCP distribution of several micrometers and OCP from submicron to several micrometer sizes. The crystallinity of a biomaterial also changes its resorption rate. Highly crystalline materials are more resistant to resorption than less crystalline materials because they are thermodynamically more stable. The crystallinity of a material can be deduced from its XRD peak. The XRD (Figure 2) patterns of the BHA and OCP samples showed broad peaks compared to the BCP sample. This is because BHA and OCP were treated at relatively lower temperatures than BCP, resulting in lower crystallinity.

Several studies have suggested possible in vivo degradation mechanisms of calcium phosphate ceramics [26,31]. Many studies have shown that biodegradable calcium phosphate materials are degraded by simple dissolution, fragmentation/disintegration, osteoclastic resorption, or phase conversion [26]. The predominant degradation mechanism will vary according to the material, and a single mechanism may not be involved. Instead, several mechanisms may be active simultaneously or sequentially. The rapid degradation of the OCP sample, which consists mainly of OCP, is believed to be due to the formation of biological apatite via fast phase conversion of OCP. In addition, OCP exhibits simple dissolution and osteoclastic resorption similar to that observed in other biodegradable calcium phosphate samples. The in vivo resorption of a grafted OCP sample is presumed to be caused by several mechanisms, including dissolution, fragmentation/disintegration, osteoclastic resorption, and phase conversion. The low crystallinity and large porosity of the OCP sample would have improved its resorption rate.

Histomorphometric analysis indicated that the OCP group produced the highest quantity of new bone formation and was the only group to show a significant difference to CON after four weeks (Figure 3). Interesting results were obtained in the OCP group related to the bone marrow. A significant amount of new bone was generated in the bone marrow after four weeks, but this almost disappeared after 12 weeks. These results suggest that OCP could influence the differentiation of stem cells into osteoblasts in the early stage [24], and it was presumed that they were resorbed over time. The cortical and medullar areas are under quite different physiological conditions. Cortical bone endures mechanical stress for bearing the body weight and movement forces, whereas the marrow area is a cavity that does not provide support against mechanical stress. Thus, bone remodeling by bone resorption and new bone formation occurs in the cortical bone and leads to the maintenance of a bone structure. On the other hand, there is no active remodeling in the marrow, which leads to the resorption of the bone structure.

The histological findings showed that the various calcium phosphate samples tested had various dissolution and new bone formation rates (Figure 5). Most calcium phosphate materials will dissolve with time after being implanted in the body, resulting in the formation of new bone in the dissolved space or on the surface of the implanted materials. The dissolution and new bone formation rates depend on the characteristics of the implanted materials, such as phase composition, porosity, Ca/P molar ratio, crystallinity, particle size, and impurities. Among the samples tested, the fastest bone healing rate was observed at four weeks in the OCP sample group. Well-organized hard callus formation was observed in both the cortical and medullary regions (Figure 5a). Most of the OCP granules were covered with new bone components, including the osteocytes of the lacuna. At 12 weeks, the bone healing process had progressed to a more mature bone form. The rapid new bone formation and bone remodeling observed in the OCP group were attributed to the rapid mineralization of the OCP crystals caused by phase conversion to biologically active apatite. On the other hand, the relatively slow new bone formation in the other test groups (BHA and OCP) was attributed to the relatively slow resorption and slow mineralization of those substances (Figure 3, Figure 4 and Figure 5). Despite the many advantages of OCP, its acid chemistry may not be a perfect combination for ideal bone substitutes, which will be addressed in a future study.

## 5. Conclusions

The OCP sample, which consisted mainly of OCP, had the fastest resorption rate in the in vivo test environment compared to the rates for BHA, which was composed mainly of low crystalline HA and MCP, which was comprised of high crystalline HA and β-TCP. The fast resorption in the OCP group was attributed to the fast phase conversion of OCP to biological apatite and to the rapid dissolution and osteoclastic resorption compared to that observed in the other biodegradable calcium phosphate test groups. The level of new bone formation in the OCP group at four weeks was much greater than those in the other test groups because of the rapid mineralization of OCP via phase conversion and the provision of space for new bone formation caused by the rapid resorption rate of OCP. In the OCP group, well-organized hard callus formation was observed after 4 weeks, and at 12 weeks, defect healing had progressed to a more mature bone form. The rapid new bone formation and bone remodeling in the OCP sample group were attributed to the rapid mineralization of OCP crystals caused by phase conversion to the rapid biologically active apatite in addition to osteoblast activation.

## Figures and Tables

**Figure 1 materials-14-05300-f001:**
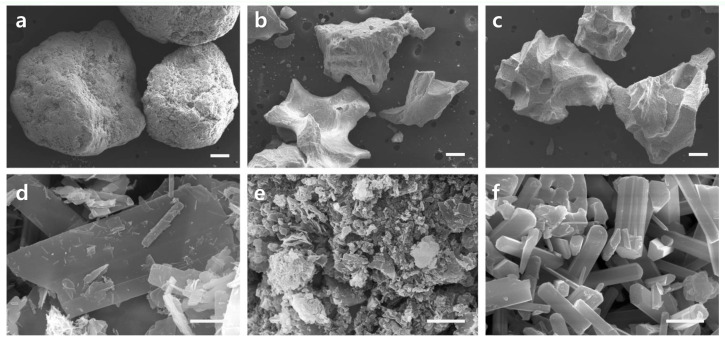
Scanning electron micrographs of OCP (**a**,**d**), BHA (**b**,**e**), and BCP (**c**,**f**) granules. The bars indicate 200 μm at lower magnification (**a**–**c**) and 1 μm at higher magnification (**d**–**f**).

**Figure 2 materials-14-05300-f002:**
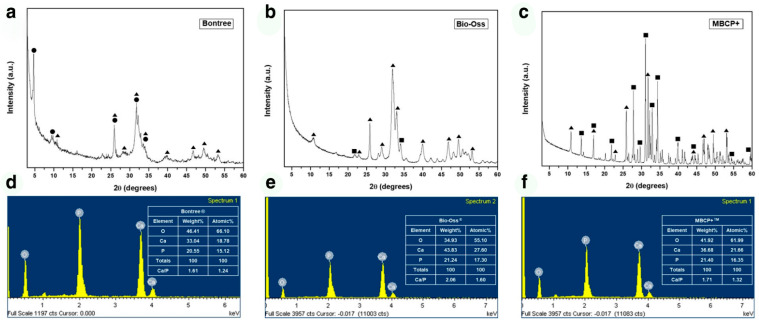
Results of XRD analysis (**a**–**c**) and EDS analysis (**d**–**f**). OCP samples (**a**,**d**) consisted of mainly OCP phase. BHA samples (**b**,**e**) consisted of HA phase, and BCP samples (**c**,**f**) consisted of HA and β-TCP phases. The OCP, HA, and β-TCP peaks are indicated by ●, ▲, and ■, respectively.

**Figure 3 materials-14-05300-f003:**
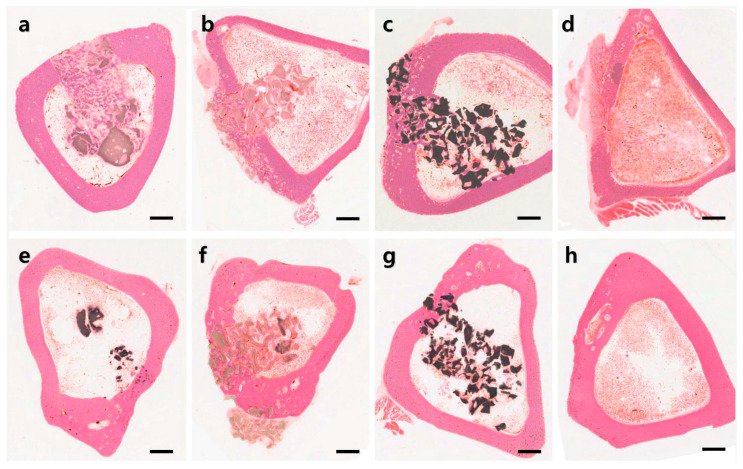
Tibias at four weeks (**a**–**d**) and 12 weeks (**e**–**h**) after implants with OCP (**a**,**e**), BHA (**b**,**f**), BCP (**c**,**g**), and CON (**d**,**h**) materials. The sample was resin embedded and ground to a 15 μm thickness. H-E stain was used. The bars indicate 2 mm.

**Figure 4 materials-14-05300-f004:**
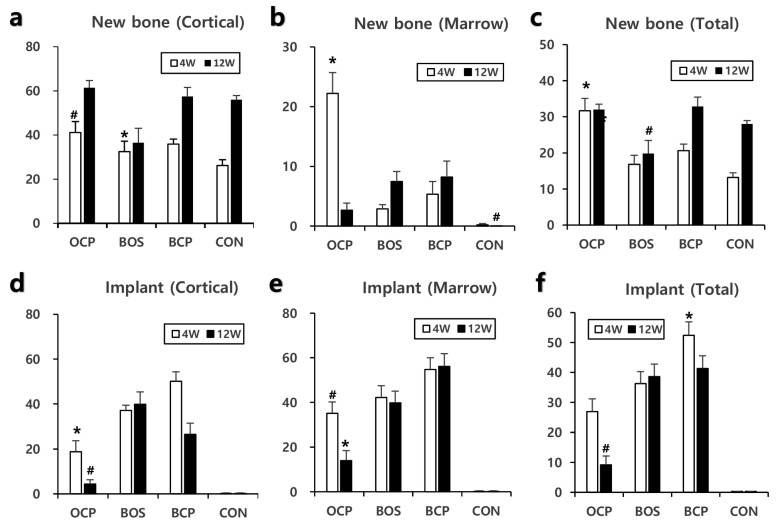
Percentages of new bone and implant remnant to the total corresponding area in cortical bone and marrow areas. The results are presented as the mean ± SE values (*n* = 9). * *p* < 0.05 between this group and other groups at same week. #: *p* < 0.05 between OCP and CON in (**a**), between CON and BHA or BCP in (**b**), between BHS and OCP or BCP in (**c**), between OCP and BHA or BCP in (**d**), between OCP and BCP or CON in (**e**), between OCP and BHA or BCP in (**f**), all in the same week.

**Figure 5 materials-14-05300-f005:**
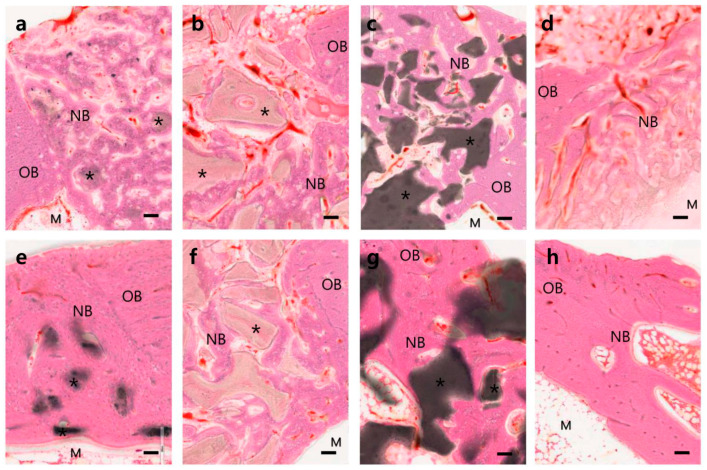
Microphotographs of the tibias at four (**a**–**d**) and 12 (**e**–**h**) weeks after implanting of OCP (**a**,**e**), BHA (**b**,**f**), BCP (**c**,**g**), and CON (**d**,**h**) materials. OB: old bone, NB: new bone, M: marrow, *: implant material. H-E stain was used; bars indicate 100 µm.

**Figure 6 materials-14-05300-f006:**
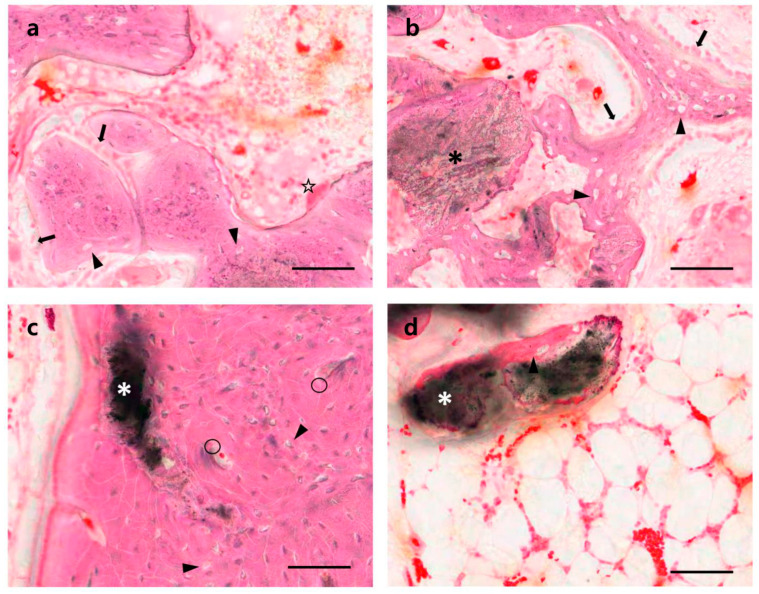
High magnification micrographs of OCP-implanted tibias. Cortical bone (**a**,**c**) and marrow area (**b**,**d**) at four (**a**,**b**) and 12 (**c**,**d**) weeks. Arrow: osteoblast, arrowhead: lacuna, *: OCP, ○: Haversian system, ☆: osteoclast. H-E stain; bars indicate 100 µm.

## Data Availability

Not applicable.

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
