# Peer review of "Biomimetic Octacalcium Phosphate Bone Has Superior Bone Regeneration Ability Compared to Xenogeneic or Synthetic Bone"

_materials, 2021, doi:10.3390/ma14185300_

Round 1

Reviewer 1 Report

  1. 1 Page 2 Line 48 “In general, the smaller the partcle size, the higher the porosity, the lower the crystallinity, and the higher the non-stoichiometric ratio, the faster -the absorption”

    What’s meaning of this sentence.

    1. Page 2 line 52 “Sintered HA bone is used as a scaffold material”

    What’s sintered HA bone?

    1. The resolution of Figure 2 is poor.

    1. It is necessary to mark the peaks of phase in the XRD pattern

    1. The legend of Figure 4 “ *; p<0.05 vs. other groups. #: p<0.05 vs CON in A, vs BHA and BCP in B, vs OCP and BCP in C, vs BHA and BCP in D, vs BCP and CON in E and vs HA and BCP in F.”

    It is very difficult to understand this description

Author Response

We appreciate your detail and kind comments. Manuscript was revised according to your suggestion and proofreading was done by an English proofreading company.

1. 1 Page 2 Line 48 “In general, the smaller the partcle size, the higher the porosity, the lower the crystallinity, and the higher the non-stoichiometric ratio, the faster -the absorption” What’s meaning of this sentence

→ Sentence was corrected clearly

2. Page 2 line 52 “Sintered HA bone is used as a scaffold material” What’s sintered HA bone?

→Sentence was changed

3. Page 4 line 152 ” it is thought that the size of the macropores formed by gathering many granules would be similar” & “the BCP microstructure was heat-treated at a higher temperature, and larger nano-sized rod particles and micropores were observed” What is the different between the micropore and macropore?

→ Sentence changed where the macropore refers to the pores formed by the gathering of several granules, and the micropore refers to the pores existing within each granule.

4. The resolution of Figure 2 is poor.

→ Figure was replaced with higher resolution image

5. It is necessary to mark the peaks of phase in the XRD pattern

→ Marks were inserted

6. The legend of Figure 4 “ *; p<0.05 vs. other groups. #: p<0.05 vs CON in A, vs BHA and BCP in B, vs OCP and BCP in C, vs BHA and BCP in D, vs BCP and CON in E and vs HA and BCP in F.” It is very difficult to understand this description

→ legend was changed easily

Reviewer 2 Report

The manuscript entitled "Biomimetic Octacalcium Phosphate Bone Has Superior Bone Regeneration Ability Than Xenogeneic or Synthetic Bone" by Jooseong Kim, Sukyoung Kim and In-Hwan Song presents the results of a small scale experimental study of a new commercial osteoplastic material based on octacalcium phosphate (OSP) in comparisom with 2 other commercial bone-substituting materials in a rabbit model of tibial cortical bone surgical defect. 

The authors attempted to prove the advantages of the OSP material over the alternative ones and vs the untreated control. They provide limited information to characterize the OSP material (XRD and EDS), and analyse the histological findings with using of the basic morphometry approach.

The study scope is relatively narrow, and this, possibly, explains the limited depth of the analysis and interpretations of the biological responses. However, while this is not a state-of-the art research, it delivers the specific messages that may be useful for the further development of the materials for bone reconstruction.

I would like to recommend to perform very careful proof-reading of the manuscript to fix non-critical but multiple grammar issues. 

A few specific critical comments to address for the improvement of the clarity and the soundness of the paper are added below:

  1. Row 2: “CON (unfilled).” Please, rephrase as "left unfilled" as control (CON)".
  2. Rows 50-51, please, correct the grammar: “Among the hydroxyapatite (HA) and ß-tricalcium phosphate (ß-TCP) materials that are currently most used as artificial bone graft materials[11, 12]. Sintered HA bone…” – the phrase is unclear.
  3. Row 56: “In a recent study, it is known” – please, correct the grammar.
  4. For the introduction in general: It is unclear, what is the motivation for the development of a new alternative calcium phosphate graft material?
  5. Row 65: “study, histological studies” – repeating words.
  6. Row 79: “those of two widely used other product types, three commercially available granular products were used.” – grammar, and repeating words…
  7. Row 91: “are synthesized at relatively low temperature” – please, explain what are advantages and limitations of this? Why this is important in the context of bone healing?
  8. Row 151-152: Please, provide the specific (quantitative) data to prove the granule size similarity between the groups.
  9. Row 157: As it is visible from SEM images at high magnification, the size distribution of the particles in the OCP material is very inhomogeneous. It's not regular in shapes and sizes, and it is much more irregular, compared with BHA and BCP.
  10. Rows 167-168: “On the other hand, the crystallinity of the OCP sample maintained the extent of crystallinity of the starting raw material because it was prepared at low temperature.” – this phrase would be more appropriate for the Discussion, not the results.
  11. Rows 171 -172: “phase, while the Bio-Oss (BHA) sample mainly consisted of the HA phase, and…” - the phrase is not completed, also there is an error in formatting.
  12. Figure 2: please, improve the figures resolution and add the interpretation of the data.
  13. Rows 179 – 182:

“material (1.33, 1.67, 1.5533).” – what does it mean?

“Ca deficient.” - Why? comparing to what?

“will be very useful when explaining the results of animal experiments in a later section” - This phrase would be more acceptable in the Discussion section.

  1. Row 193: “total defect area was significantly highest in the OCP” – grammar.

Rows 197 – 198: “This result is due to a lack of space available for bone formation because BHA does not resolve and occupy the space” - This phrase would be more acceptable in the Discussion section.

  1. Row 205: “terms of absorption rate of the” - it it absorption or resorption?
  2. Row 207 – a finding or findings?
  3. Row 215: “epithelial-lined osteoblasts” – please, correct the spelling or explain your terminology?
  4. Row 226: “a similar healing process to that of the OCP group but obvious differences were observed.” – please, correct the grammar/check the logics.
  5. In rows 280 and below, Please, specify the term “dissolution”. Is it justified to say "dissolution" in this context as the implanted material undergoes the biodegradation or bioresorption , the processes that involve various cellular populations, and it is not the purely chemical reaction of dissolution. There is some discussion on this below, but to me it does not look convincing yet.

Author Response

We appreciate your detail and kind comments. Manuscript was revised according to your suggestion and proofreading was done by an English proofreading company.

The manuscript entitled "Biomimetic Octacalcium Phosphate Bone Has Superior Bone Regeneration Ability Than Xenogeneic or Synthetic Bone" by Jooseong Kim, Sukyoung Kim and In-Hwan Song presents the results of a small scale experimental study of a new commercial osteoplastic material based on octacalcium phosphate (OSP) in comparisom with 2 other commercial bone-substituting materials in a rabbit model of tibial cortical bone surgical defect.

The authors attempted to prove the advantages of the OSP material over the alternative ones and vs the untreated control. They provide limited information to characterize the OSP material (XRD and EDS), and analyse the histological findings with using of the basic morphometry approach.

The study scope is relatively narrow, and this, possibly, explains the limited depth of the analysis and interpretations of the biological responses. However, while this is not a state-of-the art research, it delivers the specific messages that may be useful for the further development of the materials for bone reconstruction.

I would like to recommend to perform very careful proof-reading of the manuscript to fix non-critical but multiple grammar issues.

A few specific critical comments to address for the improvement of the clarity and the soundness of the paper are added below:

1. Row 2: “CON (unfilled).” Please, rephrase as "left unfilled" as control (CON)".

→Sentence was changed as suggestion

2. Rows 50-51, please, correct the grammar: “Among the hydroxyapatite (HA) and ß-tricalcium phosphate (ß-TCP) materials that are currently most used as artificial bone graft materials[11, 12]. Sintered HA bone…” – the phrase is unclear.

→Sentence was changed clearly

3. Row 56: “In a recent study, it is known” – please, correct the grammar.

→ Sentence was corrected

4. For the introduction in general: It is unclear, what is the motivation for the development of a new alternative calcium phosphate graft material?

→ So far, this has been a laboratory-scale study of OCP substances, but this study is the first large scale animal study of a commercialized OCP product. The bone regeneration ability of OCP products was compared with two of the most clinically used materials: heat-treated bovine bone and sintered BCP

5. Row 65: “study, histological studies” – repeating words.

→ Sentence was corrected

6. Row 79: “those of two widely used other product types, three commercially available granular products were used.” – grammar, and repeating words…

→ Sentence was corrected

7. Row 91: “are synthesized at relatively low temperature” – please, explain what are advantages and limitations of this? Why this is important in the context of bone healing?

→ Explain was added in the text

8. Row 151-152: Please, provide the specific (quantitative) data to prove the granule size similarity between the groups.

→ Actual size was provided

9. Row 157: As it is visible from SEM images at high magnification, the size distribution of the particles in the OCP material is very inhomogeneous. It's not regular in shapes and sizes, and it is much more irregular, compared with BHA and BCP.

→You are right. The shape of OCP granules is circular, and each OCP granule is made by aggregating ribbon-shaped OCP crystals with irregular and various crystal sizes. Here, SEM of OCP crystals is in a state without heat treatment after synthesis.

10. Rows 167-168: “On the other hand, the crystallinity of the OCP sample maintained the extent of crystallinity of the starting raw material because it was prepared at low temperature.” – this phrase would be more appropriate for the Discussion, not the results.

→Mentioned in the discussion but may be easy to understand result

11. Rows 171 -172: “phase, while the Bio-Oss (BHA) sample mainly consisted of the HA phase, and…” - the phrase is not completed, also there is an error in formatting.

→Sentence was changed clearly

12. Figure 2: please, improve the figures resolution and add the interpretation of the data.

→replaced with higher resolution image

13. Rows 179 – 182:

“material (1.33, 1.67, 1.5533).” – what does it mean?

→they are stoichiometric ratio of OCP, HA, and BCP and added explain in the sentence

“Ca deficient.” - Why? comparing to what?

→Explained in the text

“will be very useful when explaining the results of animal experiments in a later section” - This phrase would be more acceptable in the Discussion section.

14. Row 193: “total defect area was significantly highest in the OCP” – grammar.

→ sentence was corrected

Rows 197 – 198: “This result is due to a lack of space available for bone formation because BHA does not resolve and occupy the space” - This phrase would be more acceptable in the Discussion section.

→also mentioned detail in the discussion

15. Row 205: “terms of absorption rate of the” - it it absorption or resorption?

→changed into resorption

16. Row 207 – a finding or findings?

→changed into findings

17. Row 215: “epithelial-lined osteoblasts” – please, correct the spelling or explain your terminology?

→means osteoblast covered like epithelial lining, but removed to avoid confusion

18. Row 226: “a similar healing process to that of the OCP group but obvious differences were observed.” – please, correct the grammar/check the logics.

→changed sentence clearly

19. In rows 280 and below, Please, specify the term “dissolution”. Is it justified to say "dissolution" in this context as the implanted material undergoes the biodegradation or bioresorption, the processes that involve various cellular populations, and it is not the purely chemical reaction of dissolution. There is some discussion on this below, but to me it does not look convincing yet.

→Agree with you and changed in the text

Round 2

Reviewer 2 Report

The manuscript was improved as suggested by the reviewers.

Author Response

Dear reviewer

We appreciate your review. Owing to your kind and detail comment, the paper improved much. The manuscript corrected by Nurisco (certification attached) and some part was checked again by an official English simultaneous interpreter. If this version is still evaluated not suitable for Materials, we will consult once more to the company which Materials recommend.

Yours truly

In-Hwan Song
